# Towards Medical Vision-Language Contrastive Pre-training via Study-Oriented Semantic Exploration

Bo Liu
School of Computer Science,
Sichuan University,
Chengdu, China
liubo105@hotmail.com

Zexin Lu
School of Computer Science,
Sichuan University,
Chengdu, China
luzexinyyds@gmail.com

Yan Wang*
School of Computer Science,
Sichuan University,
Chengdu, China
wangyanscu@hotmail.com

## ABSTRACT

Contrastive vision-language pre-training has shown great promise in representation transfer learning and cross-modality learning in the medical field. However, without fully exploiting the intrinsic properties and correlations of multimodal medical data within patient studies, current research fails to explore all the potential of available data, leading to suboptimal performance on representation learning. In this paper, we propose a novel pre-training framework for learning better medical vision-language embedding, oriented on patients' study-level data. Based on the order-agnostic property of radiology report, we adopt a two-stage feature extraction method for more representative textual characterization. Then, by leveraging momentum encoders and memory queues, study-level semantics are explored with three contrastive objectives to provide comprehensive supervision from three perspectives, *i.e.*, cross-modal, multi-modal, and uni-modal, such that the potential information neglected by previous research can be fully exploited. The superiority of the proposed framework is demonstrated by the impressive improvements on four typical downstream tasks, including zero-shot/data-efficient image classification, image segmentation, and cross-modal retrieval.

## CCS CONCEPTS

• **Information systems** → **Multimedia and multimodal retrieval**; • **Computing methodologies** → **Unsupervised learning**; **Image representations**; **Transfer learning**.

## KEYWORDS

Medical vision-language pre-training, Vision-language alignment, Contrastive learning, Semantic representation learning.

**ACM Reference Format:**
Bo Liu, Zexin Lu, and Yan Wang. 2024. Towards Medical Vision-Language Contrastive Pre-training via Study-Oriented Semantic Exploration. In *Proceedings of the 32nd ACM International Conference on Multimedia (MM'24), October 28-November 1, 2024, Melbourne, Australia.* ACM, New York, NY, USA, 10 pages. https://doi.org/10.1145/3664647.3681531

---

*Corresponding author.

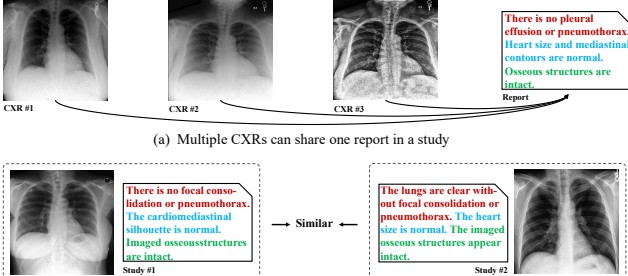

(a) Multiple CXRs can share one report in a study

(b) Different studies can share similar semantics

**Figure 1: In current patients' CXR data, there are two important but easily overlooked properties where (a) multiple CXR images can correspond to one report in a study and (b) different studies can share similar semantics.**

## 1 INTRODUCTION

The recent success of deep learning relies heavily on the large amount of annotated data. However, acquiring a sufficient amount of labeled data is particularly difficult in medical fields due to its expertise-demanding and time-consuming nature. To address this issue, many efforts [20, 28, 32, 51] have been made on vision-language contrastive pre-training to leverage the existing radiology reports. With the supervision of contrastive loss between images and their corresponding reports, the pre-trained network brings superior performance on many downstream tasks, including both multi-modal (e.g., cross-modal retrieval) and uni-modal tasks (e.g., image classification), demonstrating the effectiveness of contrastive pre-training in learning better representations on multi-modal data.

However, despite the improvement they have achieved, current methods share three weaknesses, limiting their potential. (1) Less precise data structure: in current datasets, particularly regarding chest X-ray (CXR), a collection of images can correspond to a single report, which is referred to as a patient ***study*** as shown in Figure 1 (a). Most prior methods [28, 51] directly dissociate this study into several image-report pairs whose reports are the same, resulting in false negatives during cross-modal alignment; (2) Neglected semantic similarity: unlike natural image captions, patient studies of similar disease status share many semantic similarities in terms of both images and reports (*cf.* Figure 1(b)), which are not fully exploited by previous methods; and (3) Under-explored uni-modal information: most methods focus on leveraging the cross-modal contrast as the only source of supervision for representation learning, as shown in Figure 2(a), neglecting the intrinsic information of each modality (Figure 2(c)).

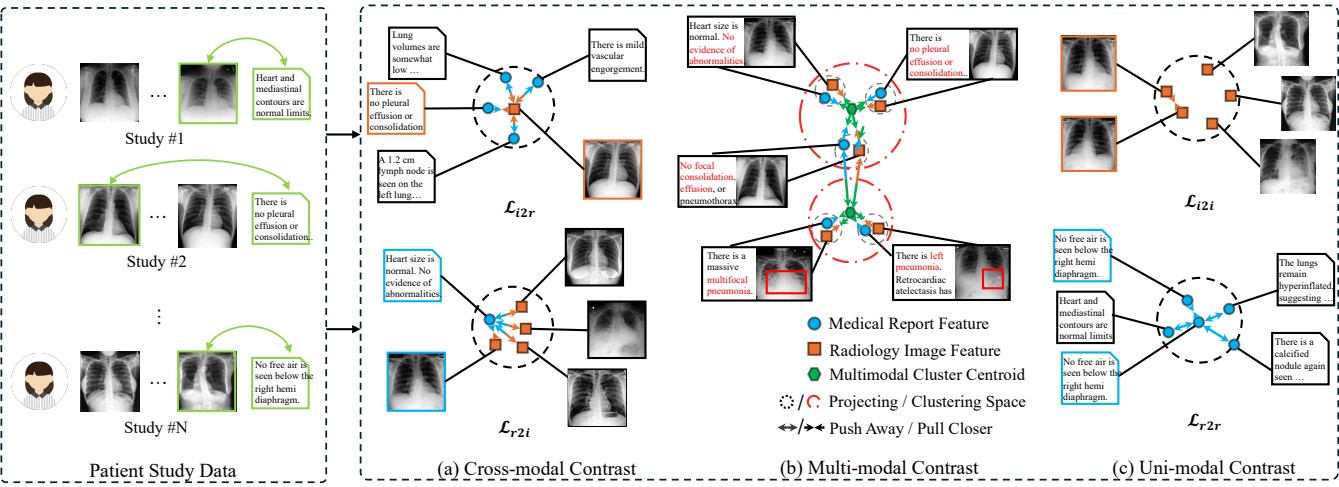

**Figure 2: Our proposed study-level contrasts for improving medical vision-language joint embeddings: (a) cross-modal, (b) multi-modal, and (c) uni-modal.**

To address the aforementioned issues, we propose a **s**tudy-ori**en**ted **s**emantic **e**xploration (SENSE) framework to improve the medical vision-language joint embedding learning via three contrastive objectives. Firstly, unlike many previous studies [20, 28, 51] which directly apply the feature extraction methods of contrastive learning for general tasks to medical field and neglect the intrinsic property of radiology reports, we adopt a two-step method for better textual feature extraction. As illustrated in Figure 4, the semantics of each sentence in a radiology report is independent, and the overall semantics of the report is irrelevant to the order of sentences. Based on such observation, we propose to encode each sentence separately and fuse them with a Max-Max method to acquire better semantic representation. Then, the momentum encoders are introduced for each modality along with three study-level supervisions (Figure 2), i.e., cross-modal, multi-modal, and uni-modal supervisions, to provide sufficient guidance for better representation learning. In particular, (1) for conducting study-level cross-modal contrast (S-CMC) (Figure 2(a)), we generate a single image-report pair for each patient study by randomly sampling one image (in case one study contains multiple ones) during each training iteration (shown in the left block of Figure 2). Then, normal instance-level contrast is performed. Moreover, with the momentum encoders and memory queues, S-CMC can sample negative pairs from the distribution of the whole dataset, effectively preventing representation collapse [16, 43]; (2) to fully exploit the semantic similarity between different studies, study-level multi-modal contrast (S-MMC) first performs K-means clustering on multi-modal study representations to encode semantic structures explicitly and enforces each image-report pair closer to its corresponding clustering centroid for semantic preservation (Figure 2(b)); (3) study-level uni-modal contrast (S-UMC) introduces additional supervision signals by conducting contrastive learning within each modality through momentum encoders and data augmentation (Figure 2(c)). Actually, with the centroid of multi-modal embeddings as the comparison target, introduced S-MMC explicitly bridges S-CMC and

S-UMC, not only further increasing the intra-modality similarities (i.e., image-to-image and report-to-report), but also encouraging the inter-modality similarities (i.e., image-to-report and report-to-image). To verify the effectiveness of the proposed method and its components, we conduct comprehensive experiments along with extensive ablation studies, covering single-modal (e.g., data-efficient image classification and segmentation) and multi-modal tasks (e.g., zero-shot image classification and cross-modal retrieval). The impressive improvements over state-of-the-art methods on all these tasks demonstrate the superiority of our framework.

The contributions of this paper can be summarized as:

- We propose a study-oriented contrastive learning framework to improve medical vision-language joint embeddings. By introducing three levels of contrastive supervision, i.e., cross-modal, multi-modal, and uni-modal, the semantics of patients' study-level data can be fully explored.
- To better exploit the multi-modal semantic similarities between different studies, we propose to encode different semantic structures through K-means clustering and force the network to learn the similarities by pushing the representations toward their corresponding centroids.
- For better characterizing the radiology reports, we propose a two-step textual feature extraction method. By first encoding sentence-wise features separately and deriving report-level representations later, order-agnostic representations can be obtained.

## 2 RELATED WORK

*Medical Image-Report Joint Learning.* In recent years, increasing studies focus on leveraging clinical reports as supervision signals to improve the learning of visual representations [5, 10, 19, 20, 28, 37, 39, 41, 44, 51]. Most methods can be categorized into two streams according to the architecture of the pre-training frameworks, i.e., encoder-decoder-based and dual-encoder-based. For the encoder-decoder-based methods [14, 44], the general paradigm is to employ

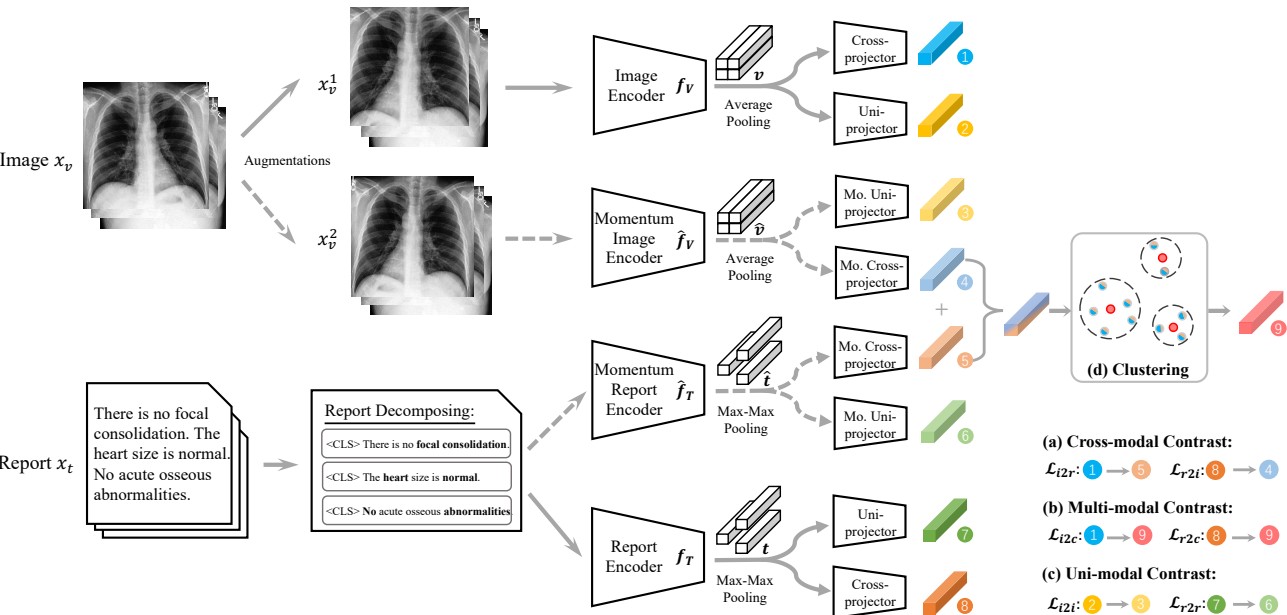

**Figure 3: Proposed SENSE framework to improve the medical vision-language joint embedding via three contrastive objectives.**

a textual decoder to translate the visual features extracted by an image encoder into reports, with an objective to minimize the variation between the generated and true reports. For the dual-encoder-based paradigm [2, 20, 28, 32, 41, 51], the pretraining objective is to align visual and textual features extracted by two separate encoders in the embedding space. Specifically, ConVIRT [51] pulls the global embeddings of paired medical image and report closer while pushing those of non-paired ones away. Following ConVIRT, BioViL [2] improves the text modeling by adding clinical vocabulary and report-specific augmentation. GLoRIA [20] and LoVT [32] further propose to conduct contrastive learning on local representations of image-report pairs to capture the fine-grained alignment in addition to the global representations. Besides, SAT [28] and MGCA [41] take similarities between different image-report pairs into consideration. However, all these methods neglect the intrinsic properties of patients' study data. We propose a study-oriented pre-training framework to explore such information fully through three introduced objectives. Recently, large multimodal models [25, 48, 52] have achieved impressive performance, especially for medical tasks [21, 30, 50]. Our pre-trained visual extractor can be incorporated by these models to better extract visual features, improving the overall performance.

*Contrastive Self-supervised Learning.* Self-supervised learning aims to endow deep neural networks with generic representations in an unsupervised manner, where contrastive learning [40] is one of the most representative paradigms. With the supervision of contrastive loss, better representation can be learned with less annotated data. For visual embedding learning, existing methods [6, 7, 16, 17, 29] utilize momentum encoder and various data augmentations for representation contrasting. In addition, to encode semantic relationships among images, [4, 27] propose to introduce

semantics-wise contrast through clustering [3, 11]. For textual embedding learning, most efforts have been made to generate highly effective positive or negative samples, such as back-translation[13, 38], token-level transformations[31, 42, 46], and semantics-level augmentations[15]. Recently, multi-modal contrastive learning [23, 26, 33, 47, 49] draws increasing attention. By pulling the paired image-text closer and pushing the non-paired ones away, better performance can be achieved for not only cross-modal tasks (e.g., cross-modal retrieval), but also uni-modal ones (e.g., image classification). However, directly applying these methods to medical tasks can hardly achieve optimal performance due to their neglect of specific properties of typical medical vision-language data. In this paper, we design a radiology-specific pre-training framework that fully exploits the semantics of patients' study-level data toward better embedding for medical images and reports.

## 3 METHODOLOGY

### 3.1 Pre-training Framework Overview

The overview of the SENSE framework is shown in Figure 3. Given an image-report pair $(x_v, x_t)$ sampled from a patient study (*cf.* Figure 2), we propose to use dual encoders for feature extraction with a visual encoder $f_V(\cdot)$ for medical image $x_v$ and a textual encoder $f_T(\cdot)$ for radiology report $x_t$, followed by a series of projectors to generate representations for dense semantic explorations, i.e., our proposed three contrastive objectives. Meanwhile, a momentum encoder is introduced in addition to each uni-modal encoder, and updated through a moving average strategy as in MoCo [17]. Particularly, the momentum image encoder is updated with $\theta_{\hat{f}_V} \leftarrow \alpha\theta_{\hat{f}_V} + (1-\alpha)\theta_{f_V}$, where $\theta$ denotes model parameters, $\hat{f}_V$ is momentum visual encoder, and $\alpha$ is a momentum coefficient.

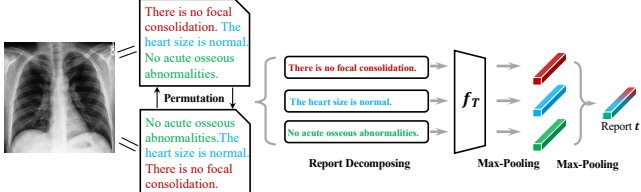

**Figure 4: The proposed Max-Max method for order-agnostic radiology report representation, motivated by the characteristic that the semantics of a report is usually invariant to the sentence order.**

Similarly, we denote momentum textual encoder as $\hat{f}_T(\cdot)$. The advantage of incorporating such dual momentum encoders for medical vision-language contrastive learning will be elaborated in the following sections. After pre-training, the robust visual and textual encoders can be readily fine-tuned for downstream tasks, including uni- and multi-modal ones.

## 3.2 Uni-modal Feature Extraction

*3.2.1 Medical Image Representation.* For the input radiology image $x_v$, a set of data augmentations, including resizing with random crop, horizontal flipping, and color jittering, which complies with the premise of preserving the semantic consistency within image-report pairs, is applied to generate two semantically correlated views $x_v^1$ and $x_v^2$. Following previous studies [2, 20, 51], we utilize the ResNet-50 pre-trained on ImageNet (before logits layer) [18] as image encoder $f_V(\cdot)$ (as well as momentum encoder $\hat{f}_V(\cdot)$), followed by an adaptive average pooling layer outputting radiology image features $v = f_V(x_v^1)$ (momentum features $\hat{v} = f_V(x_v^2)$).

*3.2.2 Radiology Report Representation.* Distinct from natural image captions, the radiology report $x_t$ has specific intrinsic properties: *it usually consists of several semantically independent sentences and its semantics is irrelevant to the order of sentences* (Figure 4). Therefore, unlike prior study [2] which embraces it with a data augmentation method by randomly permuting each sentence, we propose a more powerful order-agnostic textual feature extraction method, named **Max-Max**. Specifically, due to the presence of positional embedding in BERT-like language models, report representations gained from traditional methods [2, 20, 51] (i.e., encoding the whole report directly) are related to the order of sentences, which does not suit the radiology reports. To reduce the negative impact of sentence position feature for downstream tasks, such as cross-modal retrieval, we first decompose the radiology report into sentences and then send each sentence into the encoder individually to extract sentence-level semantics without the affection of other sentences, keeping itself intact semantically. Here, we employ BioClinicalBERT [1] as the textual encoder $f_T(\cdot)$ (momentum version $\hat{f}_T(\cdot)$) [20, 51] to extract the word-wise semantics. To obtain sentence-level representations $S = \{s_1, ..., s_N\}$, where $N$ is the number of sentences, we apply the first max-pooling layer after $f_T(\cdot)$ to all word embeddings of each sentence. Then, we derive the report-level representation $t = \text{Max-Max}(x_t)$ through the next

max-pooling layer over $S$, getting salient features for better multi-modal interaction. Similarly, we get the momentum textual features $\hat{t}$ by using our Max-Max method on momentum encoder $\hat{f}_T(\cdot)$ with the same input $x_t$. The detailed comparison between our Max-Max method and randomly sentence permutation proposed by [2] can be found in Section 4.2.1.

## 3.3 Study-oriented Semantic Exploration

*3.3.1 Study-level Cross-modal Contrast (S-CMC).* After sampling one image-report pair per patient study, we can gather a total of $N$ pairs corresponding to $N$ studies in each iteration. Considering that paired medical image and report have the highest similarity, similar to [20, 51], the S-CMC objective aims to maximize the alignment between the paired image and report (*positive pair*) while minimizing the unpaired ones (*negative pairs*) via contrastive loss [40], as shown in Figure 2(a). To address the limitation that prior studies only sample negative pairs from the current mini-batch with dozens of samples, we adopt the momentum encoder and memory queue [17, 26] to simulate the whole training dataset, which can effectively enlarge the scope of negative samples and prevent representation collapse [43]. Experimental demonstration and detailed discussion can be found in Section 4.2.1.

Specifically, we define the radiology image feature $v$ from the normal encoder as *anchor* and its paired textual feature $\hat{t}^+$ from the momentum encoder as *positive*. The $Q$ *negative* textual features $\{\hat{t}_j^-\}_{j=1}^Q$ are a dynamic set of momentum features maintained as a first-in-first-out (FIFO) *memory queue*. To achieve robust representation for downstream tasks, we follow [6] to conduct contrastive learning in the projection space, and formulate the image-to-report (*i2r*) training objective as:

$$\mathcal{L}_{i2r} = -\log \frac{\exp(\text{sim}(v, \hat{t}^+)/\tau)}{\sum_{j=1}^Q \exp(\text{sim}(v, \hat{t}_j^-)/\tau)}, \tag{1}$$

where $\text{sim}(v, \hat{t}) = \cos(p_{cro.}^v(v), \hat{p}_{cro.}^t(\hat{t}))$ with two non-linear MLP projection heads $p_{cro.}^v(\cdot)$ and $\hat{p}_{cro.}^t(\cdot)$, and cosine similarity score. $\tau$ is a temperature hyper-parameter. In practice, we update the *memory queue* using the momentum textual features from former mini-batches [17].

Similarly, as the symmetrical term of Eq. 1, we can get the report-to-image (*r2i*) contrastive loss $\mathcal{L}_{r2i}$ where $\text{sim}(t, \hat{v}) = \cos(p_{cro.}^t(t), \hat{p}_{cro.}^v(\hat{v}))$ with two non-linear MLP projection heads $p_{cro.}^t(\cdot)$ and $\hat{p}_{cro.}^v(\cdot)$ of the textual and the visual features, respectively. Thus our S-CMC loss (Figure 3 (a)) can be formulated as:

$$\mathcal{L}_{S-CMC} = \mathcal{L}_{i2r} + \mathcal{L}_{r2i}, \tag{2}$$

where a pair of report $x_t$ and image $x_v$ are aligned twice (*i.e.*, *i2r* and *r2i*) in the presence of different augmentations, providing extra supervision.

*3.3.2 Study-level Multi-modal Contrast (S-MMC).* Unlike natural image-caption datasets, the variations between different patient studies can be rather small, especially for the radiology reports of subjects with the same disease status. However, former studies did not fully explore these semantic relationships. In this section, we propose a novel method to acquire inter-study semantics explicitly and take advantage of them for extra supervision.

*Semantic Structure Mining.* Due to the difficulty in finding direct metrics to measure similarities between study data, we propose to employ unsupervised clustering to learn the semantic structure. Specifically, we first derive informative representations of a patient study from its selected image-report pair by averaging the momentum features generated from the cross-modal projectors, i.e., $\hat{m} = (\hat{p}^v_{cro.}(\hat{v}) + \hat{p}^t_{cro.}(\hat{t}))/2$, which contains the intrinsic information of both the report and image. Then, we perform K-means clustering to cluster all patient studies in the training set into $K$ groups based on $\hat{m}$ (Figure 3(d)). Note that we cluster the momentum features instead of the normal ones, as the former is expected to yield more consistent clusters during the training process for better stability. The clusters are updated once after each epoch of training.

*Semantic Structure Preservation.* The images and reports belonging to the same cluster group share large semantic relevance, which can be exploited to learn better embeddings by forcing their representations close to the cluster centroid, as shown in Figure 2 (b). Particularly, for each image, its feature $v$ and the corresponding cluster centroid $\hat{c}^+$ are adopted as a *positive* pair, while the combinations of $v$ and the rest $K-1$ cluster centroid $\{\hat{c}^-\}$ are treated as *negative* pairs. For computational efficiency, we only sample $L$ *negative* cluster centroid randomly for image-to-centroid (*i2c*) contrast in practice:

$$\mathcal{L}_{i2c} = -\log \frac{\exp(\text{sim}(v, \hat{c}^+)/\mu)}{\sum_{j=1}^{L} \exp(\text{sim}(v, \hat{c}^-_j)/\mu_j)}, \qquad (3)$$

where $\text{sim}(v, \hat{c}) = \cos(p^v_{cro.}(v), \hat{c})$, and $\mu$ is a cluster-specific scaling factor. If we use $\{\hat{m}_p\}_{p=1}^{P}$ to represent the set of feature points in each cluster, and $c^+$ to denote its centroid, following [27], the $\mu$ of a cluster can be formulated as follows to re-balance each cluster:

$$\mu = \frac{\sum_{p=1}^{P} \|\hat{m}_p - c^+\|_2}{P \log(P + \epsilon)}, \qquad (4)$$

where $\epsilon > 0$. For a loose cluster, a large $\mu$ will scale down the similarity between the images (reports) to its centroid (Eq. 3), pulling the samples belonging to the cluster closer. On the contrary, a small $\mu$ will scale up similarity, preventing the feature embeddings from collapsing into a single point and losing semantic structure.

Similarly, we have the text-to-centroid (*t2c*) training objective $\mathcal{L}_{t2c}$ for the report feature $t$ with $\text{sim}(t, \hat{c}) = \cos(p^t_{cro.}(t), \hat{c})$. The multi-modal contrast $\mathcal{L}_{S-MMC}$ (Figure 3 (b)) can be written as:

$$\mathcal{L}_{S-MMC} = \mathcal{L}_{i2c} + \mathcal{L}_{t2c}. \qquad (5)$$

3.3.3 *Study-level Uni-modal Contrast (S-UMC).* Due to the limited amount of medical image-report data, efficiently utilizing the available ones is critical for the quality of contrastive pre-training. To this end, we propose to introduce additional supervision signals within each modality via uni-modal contrast between the normal and the momentum encoders as shown in Figure 2 (c).

For visual modality, we follow [7] to align semantically correlated views $x^1_v$ and $x^2_v$, and the image-to-image (*i2i*) loss is written as:

$$\mathcal{L}_{i2i} = -\log \frac{\exp(\text{sim}(v, \hat{v}^+)/\tau)}{\sum_{j=1}^{Q} \exp(\text{sim}(v, \hat{v}^-_j)/\tau)}, \qquad (6)$$

where $\text{sim}(v, \hat{v}) = \cos(p^v_{uni.}(v), \hat{p}^v_{uni.}(\hat{v}))$ with another two non-linear MLP projection heads $p^v_{uni.}(\cdot)$ and $\hat{p}^v_{uni.}(\cdot)$. Note that unlike [47] which directly aligns $(v, \hat{v})$ from the projector heads $(p^v_{cro.}(\cdot), \hat{p}^v_{cro.}(\cdot))$ for cross-modal contrast, we introduce additional MLP projection heads for uni-modal alignment in a separate embedding space, avoiding interference with the cross-modal projection heads and preventing potential performance degradation on downstream cross-modal tasks. Naturally, we maintain another *memory queue* to store $Q$ unpaired visual momentum features.

For textual modality, we do not directly apply augmentation to the input data like the common back translation [38] or EDA methods (random insertion, random deletion, etc.) [45], due to the potential distortion of particular medical vocabularies, e.g., "cardiomediastinal silhouetter" is converted to "heart mediastinum silhouette" by back translation. Instead, we feed the same input $x_t$ to both normal and momentum encoders and conduct semantics-level augmentation by dropout mask [15]. Similar to Eq. 6, we can get our text-to-text (*t2t*) loss $\mathcal{L}_{t2t}$ with $\text{sim}(t, \hat{t}) = \cos(p^t_{uni.}(t), \hat{p}^t_{uni.}(\hat{t}))$. Finally, our S-UMC objective (Figure 3(c)) can be obtained by:

$$\mathcal{L}_{S-UMC} = \mathcal{L}_{i2i} + \mathcal{L}_{r2r}. \qquad (7)$$

Overall, we formulate our study-oriented semantic exploration loss as:

$$\mathcal{L}_{SENSE} = \lambda\mathcal{L}_{S-CMC} + \beta\mathcal{L}_{S-MMC} + \gamma\mathcal{L}_{S-UMC}, \qquad (8)$$

where $\lambda$, $\beta$ and $\gamma$ are loss balancing coefficients.

## 4 EXPERIMENTS

### 4.1 Pre-training Setup

*Dataset.* We pretrain our framework on the training set of MIMIC-CXR (version 2)[1] [24], the largest publicly available chest X-ray dataset to date. Following previous studies, we focus on the frontal view of MIMIC-CXR and leave out the other views in each patient study. Totally it contains about 146K studies for training, $1,151$ for validation, and $2,210$ for testing. In statistics, there are mainly 149,496 studies, 133,160 (89.07%) studies have only one front image, 15,596 studies have two front images (10.43%), and 740 (0.5%) studies have three or more front views. For each image, we resize the longer side to 256 pixels and use zero padding on the shorter side, making the final image size of $256 \times 256$ pixels. For each radiology report, we only keep the *Findings* section which contains detailed descriptions of the corresponding image. We preprocess all the reports by dropping special characters and symbols, such as newlines, underscores, *etc.*, and the subjects with reports of less than three words are excluded.

*Implementation Details.* All the projection heads are 2-layer MLPs with ReLU activation, outputting 512-dimensional features. For Bio-ClinicalBERT, we freeze the parameters of the first 6 layers during pre-training and only fine-tune the last 6 layers as [51]. We pretrain the whole framework for 100 epochs on 4 NVIDIA-A100 GPUs with a batch size of 128 and mixed-precision training. The network is warmed up in the first 20 epochs with a linear learning rate from $2.5 \times 10^{-6}$ to $5 \times 10^{-4}$, which further decays by cosine schedule in the remaining iterations. The optimizer is AdamW with a weight

---

[1] https://physionet.org/content/mimic-cxr/2.0.0/

**Table 1: Results of cross-modal retrieval on the test set of MIMIC-CXR [24]. Recall@$k$ (%) is used as the evaluation metrics. The best and second best results are bolded and underlined, respectively.**

| | Image-to-Report Retrieval | | | Report-to-Image Retrieval | | |
|---|---|---|---|---|---|---|
| | R@1 | R@5 | R@10 | R@1 | R@5 | R@10 |
| VSE++ [12] | 9.3 | 27.4 | 39.8 | 8.9 | 27.2 | 40.3 |
| ConVIRT [51] | 14.9 | 37.8 | 49.0 | 15.7 | 38.5 | 50.0 |
| GLoRIA [20] | 15.4 | 40.5 | 53.4 | 16.0 | 41.4 | 53.9 |
| SAT [28] | 16.5 | 42.4 | 55.3 | 17.5 | 42.6 | 55.0 |
| SENSE (Ours) - only $\mathcal{L}_{S-CMC}$ (Max) | 15.9 | 39.9 | 52.2 | 16.9 | 40.2 | 52.5 |
| SENSE (Ours) - only $\mathcal{L}_{S-CMC}$ (Max+Seq. Exc. [2]) | 16.3 | 41.4 | 52.7 | 16.8 | 40.7 | 52.1 |
| SENSE (Ours) - only $\mathcal{L}_{S-CMC}$ (Max-Max) | 16.7 | 42.1 | 53.6 | 17.2 | 42.4 | 54.1 |
| SENSE (Ours) - $\mathcal{L}_{S-CMC} + \mathcal{L}_{S-MMC}$ | 19.0 | 45.0 | 56.3 | 19.6 | 45.6 | 57.2 |
| **SENSE (Ours) - Full** | **19.5** | **45.1** | **57.3** | **19.8** | **46.2** | **57.5** |

decay of $1 \times 10^{-6}$. During each iteration, several image augmentations are applied, including random cropping with a scale from $[0.8, 1.0]$ and then resizing to $224 \times 224$ pixels, horizontal flipping with a probability of 0.5, and color jittering with brightness and contrast ranging from $[0.8, 1.3]$. Note that in order to ensure semantic consistency between images and reports, all image transformations are mild and some popular augmentations are excluded due to the potential of causing semantic inconsistency, such as Gaussian blur and excessive affine transformations, which may obscure the appearance of lesions. In terms of the hyper-parameters, we set the momentum coefficient $\alpha = 0.999$ and temperature $\tau = 0.1$ following [17], queue length $Q = 2048$, cluster number $K = 10,000$, negative cluster centroid number $L = 2048$, and loss balancing coefficient $\lambda = 1, \beta = 1, \gamma = 0.2$. For more details on implementation (even for downstream tasks), please refer to the Appendix.

*Comparison Methods.* In this section, we compare the proposed framework with four state-of-the-art medical contrastive vision-language pre-training approaches, ConVIRT [51], GLoRIA [20], BioViL [2], and SAT [28], on four common downstream tasks. **ConVIRT** is the first study focusing on contrastive vision-language pre-training in the medical domain, which simply aligns global representations of paired images and reports. **GLoRIA** proposes to additionally align local representations of image-report pairs. Based on ConVIRT, **BioViL** improves text modeling by introducing radiology-specific text encoder and augmentations. Given the same dataset used for pre-training, we can directly compare our results to the official version. **SAT** removes possible falsely negative samples by calculating the cosine similarity between reports within the mini-batch. Most baseline results are cited from **SAT** unless otherwise specified. In addition, we also compare some other methods, which will be explained in corresponding subsections.

## 4.2 Downstream Tasks

*4.2.1 Cross-modal Retrieval.* This task consists of two symmetrical parts: **Image-to-Report Retrieval** and **Report-to-Image Retrieval**. Specifically, given an image (report), the model attempts to identify the corresponding report (image). The pretrained normal image and report encoders with cross-projectors ($p^v_{cro.}(v)$ and $p^t_{cro.}(t)$) are directly used for cross-modal retrieval without further fine-tuning. We verify the performance on the official test set of

**Table 2: Test AUROC score (%) of linear classification on CheXpert [22] and RSNA [35] datasets with different portions of training data. The baseline results are cited from SAT [28] and BioViL [2].**

| Pre-training Type | Models | CheXpert | | | RSNA | | |
|---|---|---|---|---|---|---|---|
| | | 1% | 10% | 100% | 1% | 10% | 100% |
| Contrastive | Random | 56.1 | 62.6 | 65.7 | 58.9 | 69.4 | 74.1 |
| | ImageNet | 74.4 | 79.1 | 81.4 | 74.9 | 74.5 | 76.3 |
| | VSE++ | 50.3 | 51.2 | 52.4 | 49.4 | 57.2 | 67.9 |
| | ConVIRT | 85.7 | 87.0 | 87.5 | 85.4 | 87.4 | 88.0 |
| | GLoRIA | 86.3 | 87.9 | 88.2 | 86.2 | 87.6 | 88.9 |
| | BioViL | - | - | - | **88.1** | 88.4 | 89.1 |
| | SAT | 86.9 | 88.3 | 88.6 | 87.4 | 89.2 | 90.2 |
| | SENSE - 25% | 86.1 | 86.9 | 87.9 | 86.7 | 88.6 | 90.1 |
| | SENSE - 50% | 87.2 | 88.0 | 89.2 | 87.1 | 89.0 | 90.5 |
| | SENSE - 75% | 87.3 | 88.1 | 89.2 | 87.3 | 89.1 | **90.7** |
| | **SENSE - 100%** | **87.6** | **88.5** | **89.3** | 88.0 | **89.6** | **90.7** |
| Masked | PTUnifier | 88.7 | 89.0 | 90.1 | - | - | - |
| | M3AE | 84.0 | 86.4 | 88.9 | 86.7 | 88.0 | 89.5 |

**MIMIC-CXR** [24] with a total of 2461 image-report pairs (i.e., there are multiple pairs with the same report) and use Recall@$k$ as evaluation metrics, where $k = 1, 5, 10$. Besides, the result of VSE++ [12] is also presented for its outstanding performance on cross-modal retrieval task in the general domain.

From the results in Table 1, we have the following observations. (1) The first variant of our method ('only $\mathcal{L}_{S-CMC}$ (Max)') can be considered as an extended version of ConVIRT enhanced by the dual momentum encoders. Its superior performance to the original ConVIRT demonstrates the effectiveness of the dual momentum encoder architecture for medical contrastive vision-language pre-training. (2) The proposed Max-Max strategy can better extract textual representations for the retrieval tasks, leading to improved performance compared to single Max pooling and sequence exchange (denoted as Seq. Exc.) data augmentation [2] on all evaluations. (3) When further incorporating the semantic similarities through the clustering-based S-MMC, we achieve significant improvements on both retrieval tasks (at least $\sim 2.3\%$ on all metrics), and substantially outperform all existing methods. (4) When lastly

**Table 3: Test results (%) of RSNA Pneumonia [35] zero-shot classification task. Accuracy and F1 score are reported.**

| Metrics | ConVIRT | GLoRIA | SAT | BioViL | **SENSE** |
|---------|---------|--------|-----|--------|-----------|
| Accuracy | 66.6 | 70.8 | 72.4 | 73.2 | **74.1** |
| F1 Macro | 61.6 | 56.7 | 62.0 | 66.5 | **67.3** |

adding the uni-modal contrast to compose our full model, the optimal performance is achieved on both retrieval tasks and all metrics.

*4.2.2 Data-efficient Image Classification.* To evaluate the data efficiency of our framework, we pre-train the network with different amounts of data (i.e., 25%, 50%, 75%, and 100%). Following the classification protocol in previous studies, we freeze the weights of the pre-trained image encoder and fine-tune a randomly initialized linear classifier with 1%, 10%, and 100% training data to evaluate the effectiveness of visual representation learning. We conduct classification on two datasets: (1) **CheXpert** is a multi-label chest X-ray dataset where each image is labeled based on the appearance of 14 kinds of disease symptoms. We follow the same experimental settings in [20, 28, 51] to use the validation set for evaluation because the original test set has not been made publicly available. Moreover, we randomly pick 5, 000 images from the training set for validation; (2) **RSNA Pneumonia** consists of two types of chest X-ray images, *i.e.*, health and pneumonia. We use the same preprocessing procedure and experimental setting as in [20, 28]. Specifically, about 30,000 front view images are split into training/validation/test sets with a ratio of 70%/15%/15%. For both datasets, we resize the image to $256 \times 256$ pixels

The area under the receiver operating characteristic curve (AUROC) results are displayed in Table 2. The results suggest that our method achieves state-of-the-art results on both datasets in most settings (only 0.1% lower than BioViL on **RSNA** with 1% data). Remarkably, with only 50% pre-training data, our method has already achieved comparable results with the previous best results except for BioViL on **RSNA** 1% data setting, demonstrating the excellent data efficiency characteristics of our framework. Besides the comparison with methods in contrastive paradigm, we also compare two representative methods in masked pre-training, PTUnifier [8] and M3AE [9]. It shows that our SENSE can achieve higher or comparable results. The reason why our SENSE is slightly lower than PTUnifier [8] may be because PTUnifier [8] used more pre-training data.

*4.2.3 Zero-shot Image Classification.* Inspired by CLIP [33], we treat the zero-shot image classification task as identifying the most similar description among a set of class-specific prompt descriptions for a given image. The pre-trained normal image and report encoders with cross-projectors ($p_{cro.}^v(v)$ and $p_{cro.}^t(t)$) are directly used for zero-shot image classification. Following the settings of BioViL [2], which achieves the best performance among previous studies, we evaluate the model's zero-shot recognition ability on the test set of **RSNA Pneumonia** [35], where class prompts for the healthy and unhealthy are set as "no evidence of pneumonia" and "findings suggesting pneumonia", respectively.

**Table 4: Test Dice score (%) of pneumothorax segmentation on the SIIM[36] dataset with different portions of training data.**

| Initialization Method | Pneumothorax Segmentation | | |
|-----------------------|------|------|------|
| | 1% | 10% | 100% |
| Random | 14.5 | 48.7 | 58.8 |
| ImageNet | 49.7 | 62.6 | 73.3 |
| ConVIRT [51] | 57.0 | 67.4 | 72.3 |
| GLoRIA [20] | 58.1 | 67.5 | 73.1 |
| SAT | 59.2 | 68.2 | 74.7 |
| **SENSE (Ours)** | **60.0** | **69.5** | **75.2** |

**Table 5: Results of image-to-report retrieval on the test set of MIMIC-CXR [24] with different report feature extraction methods. Recall@$k$ (%) is used here as the evaluation metric.**

| Base method | Image-to-Report Retrieval | | |
|-------------|------|------|------|
| SENSE - only $\mathcal{L}_{S-CMC}$ | R@1 | R@5 | R@10 |
| *One-step:* | | | |
| + Mean Pooling | 14.8 | 38.0 | 48.0 |
| + Token Pooling | 14.9 | 37.7 | 49.3 |
| + Max Pooling | **15.9** | **39.9** | **52.2** |
| *Two-step:* | | | |
| + Max-Mean Pooling | 15.7 | 39.3 | 50.7 |
| + Max-Max Pooling | **16.7** | **42.1** | **53.6** |

As presented in Table 3, our method achieves the best results on both accuracy and F1 metrics, demonstrating that the learned joint embeddings are effective for zero-shot classification.

*4.2.4 Medical Image Segmentation.* Following SAT [28], we utilize the pre-trained visual encoder as the encoder for U-Net [34] and fine-tune the whole network with the supervision of annotated pathological chest X-ray images under 1%, 10%, and 100% training data. The dataset used in this task is **SIIM-ACR Pneumothorax** [36], which contains 12, 047 chest radiographs and is split into training/validation/test sets in a ratio of 70%/15%/15%.

We report the Dice score for evaluation in Table 4. The results show that the network pre-trained with our SENSE is more effective than the others in all settings, especially at the 1% and 10% training data settings, demonstrating the superiority of our pretraining framework.

## 4.3 Analysis and Ablation Study

In this part, we conduct ablation studies and analyses to verify the effectiveness of each component of the framework. More experiments can be found in Appendix.

*4.3.1 Effect of Different Report Feature Extraction Methods.* Because the output of the textual encoder (i.e., BioClinicalBERT [1]) is a feature matrix consisting of the representations of all tokens, we evaluate the performance of different pooling methods to obtain report representations. To avoid the interference of other factors, we only use the $\mathcal{L}_{S-CMC}$ objective here because it has the most direct impact on the cross-modal retrieval task. As displayed in Table 5,

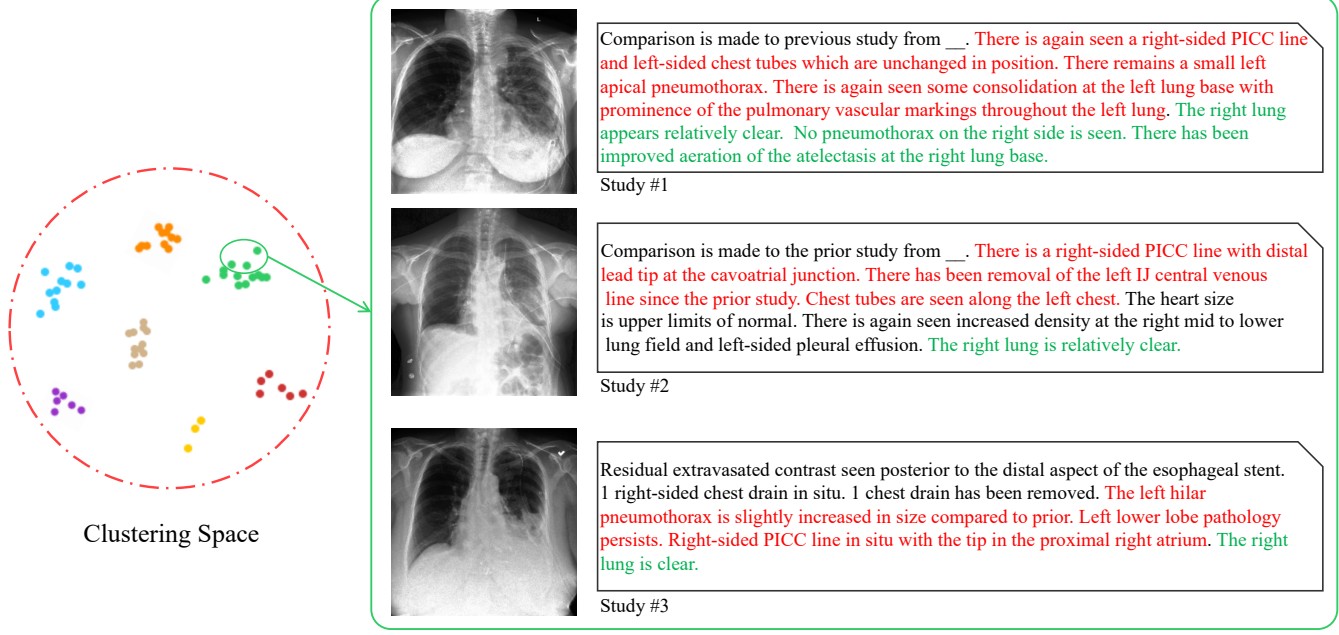

**Figure 5: Example patient studies with contained image-report pair in MIMIC-CXR [24] sampled from clusters generated by the proposed SENSE (in terms of $\mathcal{L}_{S-MMC}$). The texts in the same color describe similar findings.**

**Table 6: Test AUROC score (%) of linear image classification (100% pre-training data) on CheXpert [22] and RSNA [35] datasets with various combinations of SENSE pre-training objectives.**

| Pre-training Objectives | | | CheXpert | | | RSNA | | |
|---|---|---|---|---|---|---|---|---|
| $\mathcal{L}_{S-CMC}$ | $\mathcal{L}_{S-UMC}$ | $\mathcal{L}_{S-MMC}$ | 1% | 10% | 100% | 1% | 10% | 100% |
| ✓ | | | 86.3 | 87.3 | 88.0 | 87.0 | 88.3 | 90.1 |
| ✓ | ✓ | | 87.0 | 87.7 | 88.8 | 87.3 | 88.8 | 90.0 |
| ✓ | | ✓ | 87.2 | 88.1 | 88.9 | 87.6 | 89.1 | 90.5 |
| ✓ | ✓ | ✓ | **87.6** | **88.5** | **89.3** | **88.0** | **89.6** | **90.7** |

max pooling can perform better than the others in the one-step method, and the proposed two-step method Max-Max achieves the best results on all three metrics.

*4.3.2 Impact of Different Pre-training Objectives.* As we have explored the impact of different pre-training objectives on cross-modal retrieval in Section 4.2.1, here we investigate the impact on the transfer learning task (e.g., image classification). As reported in Table 6, both $\mathcal{L}_{S-UMC}$ and $\mathcal{L}_{S-MMC}$ could enhance visual representations, demonstrating that the introduced uni-modal and multi-modal supervisions can effectively supplement the cross-modal interactions.[2] When combining all of $\mathcal{L}_{S-CMC}$, $\mathcal{L}_{S-UMC}$ and $\mathcal{L}_{S-MMC}$ as our proposed SENSE, optimal performance can be achieved.

---

[2]As the foundation for contrastive vision-language pre-training, the cross-modal contrast ($\mathcal{L}_{S-CMC}$) cannot be ablated.

## 4.4 Qualitative Evaluation

To gain insight into the qualitative understanding of what our model has learned during training, particularly with the incorporation of $\mathcal{L}_{S-MMC}$, we present pairs of images and corresponding reports from each patient study in MIMIC-CXR [24], sampled from one cluster as depicted in Figure 5. It can be seen that the images (reports) belonging to the same cluster present (describe) highly similar symptoms although they do not belong to the same patient study, demonstrating that our SENSE does capture meaningful semantics structure with the proposed study-level clustering.

## 5 CONCLUSION

In this paper, we proposed a study-oriented pre-training framework to improve medical vision-language joint embeddings. Leveraging the order-agnostic property of radiology report, a two-step feature extraction method was adopted to obtain better texture representations. Moreover, through the momentum encoders and memory queues, three patients' study-level contrastive objectives, including uni-modal, cross-modal, and multi-modal were introduced to provide sufficient guidance for semantics exploring from limited data. Extensive experiments on four downstream tasks, covering single-modal and multi-modal tasks, demonstrated the superiority of the proposed framework for better medical vision-language representation learning.

*Limitations and Future Work.* This work mainly focuses on visual representation learning and ignores validating the effectiveness of textual ones, which can be seen as a limitation. More experiments about the performance of textual encoder on downstream tasks, such as report classification, will be conducted in future work.

# ACKNOWLEDGMENTS

We thank all reviewers for their valuable time and feedback. This work is supported by National Natural Science Foundation of China (NSFC 62371325, 62071314), Sichuan Science and Technology Program 2023YFG0025.

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
