# OpenReview forum: "Towards Medical Vision-Language Contrastive Pre-training via Study-Oriented Semantic Exploration"
_acmmm.org/ACMMM/2024/Conference — MM2024 Poster_

### Official Review · Reviewer_ZWax · 2024-05-01

**Rating:** 5
**Confidence:** 4

**Summary:**

The proposed paper introduces a novel pre-training framework aimed at enhancing medical vision-language embeddings by utilizing patient study-level data, addressing the limitations of current research in fully capitalizing on multimodal medical data. It employs a two-stage feature extraction process for textual data, momentum encoders, and memory queues to explore study-level semantics through three contrastive objectives: cross-modal, multi-modal, and uni-modal. Demonstrated improvements in downstream tasks like image classification, segmentation, and cross-modal retrieval, alongside comprehensive ablation studies, underscore the effectiveness of the framework, with plans to release the code and model upon acceptance.

**Strengths:**

Enhanced Data Utilization: The framework effectively leverages patient study-level data, which is a rich source of information often underutilized in existing models. By focusing on these comprehensive datasets, the model can capture deeper, more nuanced correlations and intrinsic properties across different modalities, leading to a more robust understanding and representation of medical data.

Innovative Feature Extraction and Learning Techniques: Utilizing a two-stage feature extraction method acknowledges and harnesses the order-agnostic nature of radiology reports for more accurate textual characterization. Additionally, the integration of momentum encoders and memory queues to explore study-level semantics through diverse contrastive objectives (cross-modal, multi-modal, and uni-modal) enables the model to capture a broad spectrum of data characteristics and relationships, enhancing the learning process.

Proven Effectiveness and Comprehensive Evaluation: The framework's superiority is demonstrated through significant performance improvements across multiple downstream tasks, including zero-shot/data-efficient image classification, image segmentation, and cross-modal retrieval. This broad array of tasks showcases the model's versatility and effectiveness in practical applications. Furthermore, the inclusion of comprehensive ablation studies provides a thorough validation of each component's contribution to the overall performance, emphasizing the thoughtfulness and rigor of the research design.

**Limitations:**

Insufficient Comparison with Relevant Literature: The paper does not sufficiently compare its approach with other significant works in the domain of medical vision-and-language pre-training. Specifically, it lacks a comparative analysis with recent and relevant methodologies such as "Multi-Modal Masked Autoencoders for Medical Vision-and-Language Pre-Training," "Towards Unifying Medical Vision-and-Language Pre-training via Soft Prompts," and "Mapping medical image-text to a joint space via masked modeling." This omission is notable because these works also explore innovative ways to integrate and leverage multimodal data for improved representation learning.

**Suitability:**

3

---

### Official Review · Reviewer_W3RE · 2024-05-22

**Rating:** 4
**Confidence:** 3

**Summary:**

The authors propose a study-oriented contrastive learning architecture to learn vision-language correspondence, which adopts multi-level contrastive learning methods to model the cross-modal alignment and boost multiple downstream tasks. Experimental results show the effectiveness of the model. However, there are some minor concerns.

**Strengths:**

This paper is well-written, with clear figures and descriptions that make it easy for readers to understand the ideas, methods, and results.
The authors have conducted necessary experiments to demonstrate the effectiveness of modeling medical vision-language alignment.

**Limitations:**

1. In Section 3.2.2, using max pooling twice to learn report representations is quite confusing. Generally, pooling operation may lead to a significant loss of semantic information, such as sentence structure features. Additionally, it may also weaken the significance of key pathological words in sentences. These drawbacks seem contrary to the paper's motivation (e.g. learning intrinsic
properties of medical data). Why not apply other methods to build report representations, such as a simple MLP?

2. Is the choice of loss coefficients in Eq.8 based on experience? It is recommended to include parameter experiments to quantitatively show how changes in the coefficients affect the final results. This would provide a more comprehensive understanding of the impact of each contrastive learning component.

**Suitability:**

3

---

### Official Review · Reviewer_k3LB · 2024-05-24

**Rating:** 4
**Confidence:** 3

**Summary:**

This paper introduces a novel pre-training framework called SENSE, designed to enhance medical vision-language joint embeddings through study-oriented semantic exploration. It employs a two-step feature extraction method to better capture the intrinsic properties of radiology reports and leverages momentum encoders and memory queues to introduce three contrastive objectives at the study level—cross-modal, multi-modal, and uni-modal—to thoroughly exploit the semantic information within patient data.

**Strengths:**

This paper proposes a new method to fully explore the intrinsic properties and correlations of multimodal medical data in datasets, which to some extent solves the problem of poor representation learning performance.

This paper demonstrates the effectiveness of the proposed method through extensive and sufficient experiments on the MIMIC-CXR dataset and multiple downstream tasks.

**Limitations:**

This paper does not raise novel questions, and the contribution of the paper is not prominent enough. Moreover, the innovation of the proposed method is not strong enough, which is similar to the following works：
[1]Multi-Granularity Cross-modal Alignment for Generalized Medical Visual Representation Learning.
[2]GLoRIA- A Multimodal Global-Local Representation Learning Framework for Label-efficient Medical Image Recognition.

This paper mentions one of the weaknesses of the current method in lines 90-96, which states that "in current datasets, particularly regarding chest X-ray (CXR), a collection of images can correspond to a single report". However, in chest X-ray datasets such as the MIMIC-CXR dataset, an imaging study of a patient generally includes two images: a frontal view and a lateral view, corresponding to one report. The author should explain this issue.

This paper only conducted experiments on the MIMIC-CXR dataset, and the author should conduct experiments on another common chest X-ray dataset, IU X-Ray, to verify the effectiveness and generalization of the proposed method.

**Suitability:**

3

---

### Official Review · Reviewer_Vb2T · 2024-05-24

**Rating:** 5
**Confidence:** 3

**Summary:**

This paper focuses on the current challenges faced by visual language models in the field of medicine. These challenges include the underutilization of the intrinsic properties and correlations present in multimodal medical data, as well as fully harnessing the potential of the available data. To overcome these challenges, the paper proposes a study-oriented contrastive learning framework. By incorporating three levels of contrastive supervision, including cross-modal, multi-modal, and uni-modal, the framework enables a comprehensive exploration of the semantics within patient research-level data. The proposed approach has been extensively validated across a wide range of downstream tasks, yielding promising results.

**Strengths:**

1. The article focuses on special and specific challenges of the vision language model faced in the medical field: scarcity and unstructured data. The paper fully uses the medical data across three levels: cross-modal, multi-modal and uni-modal.

2. Extensive experiments demonstrate that the proposed contrastive learning pre-training is effective on multiple downstream tasks, including retrieval, classification, zero-shot classification and segmentation. The experiments are detailed and sufficient.

3. The paper is well written and organized.

**Limitations:**

Related works on vision language large models in the medical field should be summarized, such as [1].

[1] @misc{yang2023enhancing,
    title={Enhancing Visual Grounding and Generalization: A Multi-Task Cycle Training Approach for Vision-Language Models},
    author={Xiaoyu Yang and Lijian Xu and Hao Sun and Hongsheng Li and Shaoting Zhang},
    year={2023},
    eprint={2311.12327},
    archivePrefix={arXiv},
    primaryClass={cs.CV}
}

**Suitability:**

3

---

### Meta-Review · Area_Chair_TzBd · 2024-06-30

**Recommendation:** Accept (Poster)
**Confidence:** 4

**Metareview:**

All reviewers appreciate the novelty and technical contributions of this work. However, they point out that the experimental comparisons are insufficient and require further improvement. Based on the overall ratings, I suggest accepting this work. The authors are recommended to improve the manuscript based on the feedback before submitting the final version.